# Use of the Barthel Index to Assess Activities of Daily Living before and after SARS-COVID 19 Infection of Institutionalized Nursing Home Patients

**DOI:** 10.3390/ijerph18147258

**Published:** 2021-07-07

**Authors:** Bibiana Trevissón-Redondo, Daniel López-López, Eduardo Pérez-Boal, Pilar Marqués-Sánchez, Cristina Liébana-Presa, Emmanuel Navarro-Flores, Raquel Jiménez-Fernández, Inmaculada Corral-Liria, Marta Losa-Iglesias, Ricardo Becerro-de-Bengoa-Vallejo

**Affiliations:** 1SALBIS Research Group, Faculty of Health Sciences, Universidad de León, 24071 León, Spain; btrer@unileon.es (B.T.-R.); pilar.marques@unileon.es (P.M.-S.); cliep@unileon.es (C.L.-P.); 2Research, Health and Podiatry Unit, Department of Health Sciences, Faculty of Nursing and Podiatry, Universidade da Coruña, 15403 Ferrol, Spain; daniellopez@udc.es; 3Faculty of Health Sciences, Universidad de León, 24071 León, Spain; epereb@unileon.es; 4Frailty Research Organizaded Group (FROG), Department of Nursing, Faculty of Nursing and Podiatry, University of Valencia, 46010 Valencia, Spain; emmanuel.navarro@uv.es; 5Faculty of Health Sciences, Universidad Rey Juan Carlos, 28933 Madrid, Spain; raquel.jimenez@urjc.es (R.J.-F.); inmaculada.corral.liria@urjc.es (I.C.-L.); 6Facultad de Enfermería, Fisioterapia y Podología, Universidad Complutense de Madrid, 28040 Madrid, Spain; ribebeva@ucm.es

**Keywords:** activities of daily living, Barthel index, SARS-CoV-2

## Abstract

The objective of the present study was to evaluate the activities of daily living (ADLs) using the Barthel Index before and after infection with the severe acute respiratory syndrome coronavirus 2 (SARS-CoV-2) and also to determine whether or not the results varied according to gender. The ADLs of 68 cohabiting geriatric patients, 34 men and 34 women, in two nursing homes were measured before and after SARS-CoV-2 (Coronavirus 2019 (COVID-19)) infection. COVID-19 infection was found to affect the performance of ADLs in institutionalized elderly in nursing homes, especially in the more elderly subjects, regardless of sex. The COVID-19 pandemic, in addition to having claimed many victims, especially in the elderly population, has led to a reduction in the abilities of these people to perform their ADLs and caused considerable worsening of their quality of life even after recovering from the disease.

## 1. Introduction

Coronavirus 2 (severe acute respiratory syndrome coronavirus 2 (SARS-CoV-2)) is an infectious disease that causes a severe acute respiratory syndrome. This virus belongs to the family of positive-sense enveloped RNA beta coronaviruses that emerged in Wuhan, China, in December 2019 [1]. It is the cause of the clinical disease known as COVID-19, which has caused more than 50 million infections and more than 1.25 million deaths according to the World Health Organization (WHO) [2].

In Spain, 5417 nursing homes exist, with 690 of these in Castilla y León, of which 71% are private [3].

According to the recent monographic report on Spain from the ltccovid.org portal, a site that belongs to the International Long-Term Care Policy Network, which is a network managed by the London School of Economics (LSE), data updated on May 28 indicate that 237,906 people have been infected by COVID-19 in Spain and 27,119 have died from this disease. Deaths in nursing homes have risen to 19,194, which is 70% of the total number of deaths, with 2449 in Castilla y León [4]. 

In a disease as infectious as COVID-19, host factors are the key to determining the severity and progression of the disease [5]. For severe COVID-19 disease, the main risk factors include age, male gender, obesity, smoking, and comorbid chronic diseases, such as hypertension, type 2 diabetes mellitus, and others [6,7,8].

Symptoms that are presented by COVID-19-infected patients vary from one person to another and can mimic symptoms present in other common infections. The most frequently found symptoms are fever, cough, myalgia, fatigue, dyspnea, anosmia, and ageusia [9,10]. Sometimes, these patients report having increased sputum production, headache, hemoptysis, diarrhea, and myalgia [11,12,13,14,15,16], although it is believed that approximately 20% of patients do not present any type of symptom [17]. The average recovery time in someone with mild illness is two weeks, while in severe illness, it can be 3–6 weeks [18]. After several weeks of convalescence, rehabilitation is essential to recover functionality as soon as possible, especially in the elderly population. The goal of rehabilitation in patients with COVID-19 infection is to facilitate improvements in the sensation of dyspnea, relieve anxiety and depression, reduce virus-associated complications, improve functionality, preserve pre-existing functions, and improve the quality of life by helping these patients to regain the same level of physical and functional independence that they had before contracting the disease.

After resolution of the acute phase, physical, emotional, and psychological impairments can often persist for a prolonged period and contribute to complex and multi-factorial disabilities that require continuous care and rehabilitative multimodal management [19,20,21]. As part of the rehabilitative therapy of these patients, evaluation of several factors that can affect these individuals once they have recovered from the virus is recommended: (1) deterioration of general functionality, (2) deterioration in the ability to carry out activities of daily living (ADLs), and (3) social disadvantages, which are evaluated using scales such as the Performance Status, the Barthel index, and the Functional Independence Measure [22].

In a community dwelling, screening of older people and assessing their abilities to conduct ADLs, such as getting out of bed, toileting, bathing, dressing, grooming, and eating, are frequently used as indicators of the functional status of an individual. These measures are applied to detect early onset of disability and are key factors for care management [23].

Geriatric assessment using the Barthel index is very important when aiming to optimize the care of elderly patients during a new epidemic outbreak. Therefore, the goal of this study was to evaluate ADLs in residents before and after contracting COVID-19 by establishing a period of time in which no underlying disease or prompt rehabilitation could invalidate the results. The Barthel index was selected as the assessment tool to verify if an individual’s ADLs decreased after overcoming the infection, which activities were most affected, and whether or not the gender of the elderly person affected the results.

## 2. Materials and Methods

The clinical study was designed as a longitudinal prospective cohort study. This study was approved by the ethics committee of the San Carlos Clinical Hospital of Madrid with internal code 21/251-E. Authorization was requested from the management of the two geriatric residences under study, and all patients provided informed consent before enrolling in the study. From March 2020 to December 2020, 68 residents contracted and overcame the SARS-CoV-2 virus, with different degrees and affectations.

For the sample size, we based our calculations on previous results obtained by Masanori Okamoto et al. [24] in which they analyzed the Barthel index in patients who had been diagnosed with benign tumors and then underwent surgery. They then compared these patients to the patients diagnosed with atypical lipomatous tumors who were surgically treated and obtained results of 98.01 ± 0.62 and 97.08 ± 2.49, respectively. For a two-tailed test with an α level of 0.05 and 95% confidence interval (CI) and a statistical analysis of the desired power of 80% (error β = 20%), a minimum sample size of 59 people was obtained, and after estimating a 15% dropout rate, a total of 68 people was needed.

The ADLs were evaluated in the residents periodically by the nurse using the Barthel index. The Barthel index (or Barthel scale) is an instrument used in medicine for the functional assessment of a patient.

In the homes for the elderly in Spain, more specifically in those of Castilla y León, the community in which our study was conducted, the ADLs of residents were evaluated in order to establish the degree of dependency, a scale established by Royal Decree 504/2007 of 20 April for determining the dependency situation established by the Law 39/2006, of 14 December. This decree addresses the promotion of personal autonomy and care for people who are in a dependent situation. Based on this decree, it is mandatory for nursing homes to conduct at least two determinations of the Barthel index in these types of residences annually; therefore, based on this regional law and the Barthel Index, which has a track record of being used in numerous studies and is still widely used today as a simple method to assess ADL for various diseases [25], we decided that this index would be a good indicator for evaluating the impact that the pandemic had had on our institutionalized elders.

This scale is used to measure the ability of a person to perform 10 basic ADLs; in this way, a quantitative estimate of their degree of independence can be obtained. The scale is also known as the Maryland Disability Index. The patient is questioned with respect to different activities, and their abilities to perform each of the corresponding activities is assigned a score according to their ability to perform the activity.

In the case of washing and grooming, if a patient can perform it without any complication, a maximum score of 5 points is given; however, if a patient cannot perform the activity, they are assigned a score of 0. The activities of eating, dressing, stooling, urinating, using the toilet, and climbing steps can have maximum score of 10 points if an individual can successfully perform the activity; however, if they need help with the activity, 5 points are given; if they are unable to perform it, a score of 0 is assigned. Finally, moving and walking carry maximum scores of 15 points each. If a person needs a minimal amount of help to carry out the activity, the person receives a score of 10 points, whereas if the person needs more help, they will be awarded 5 points. If the person is completely dependent on help, a score of 0 is assigned.

Once all of the scores are obtained, the sum is organized in such a way that the totally independent residents will have a score of 100, the residents with mild dependency will have a score of 91 to 99 points, moderate dependency is established with a score of 61 to 90, severe dependence entails scores ranging from 21 to 60 points, and total dependence is considered as a score of 20 points or less [25,26]. The Barthel index values obtained at a maximum of three months before the disease was contracted were used as the reference values. Indices were obtained at a maximum of three months after overcoming the infection and being discharged by the medical team in order to assess whether the infection had changed the status of the ADLs and therefore the degree of dependence had changed. 

All of the individual Barthel scores were collected in an Excel table after scores were collected in the morning to ensure that the tiredness derived from the daily activity was not a factor that would have altered the results.

All of the results were added to determine the score by item and the total score.

The facilities used in the sample are among those with the greatest number of elderly people, and they are the residences that have more control over the ADLs. Among these places, some of them are arranged (indicating that part of the care derived from living in the residence receives public funding) in which the administration performs a very exhaustive monitoring of the places. This type of residence conducts Barthel assessments every three months; thus, when the pandemic affected these facilities, all of the elderly residents had been tested with the index, at most, three months prior to the start of the pandemic. After taking into account that SARS-CoV-2 was contracted by residents in the facility, affecting almost all them in a short period of time, we were able to obtain the Barthel index values in the survivors immediately after contracting the virus up to a maximum post-exposure time of three months, thus assessing the pure effects of the virus without the possibility of improvement after rehabilitation, derived diseases, and complications.

The ability to eat in the dining room of the residence hall or in the resident’s room was evaluated, the ability to dress was assessed in the resident’s room and always with their usual clothes, and the ability to go to the toilet and pass urine and stool was evaluated in their own toilet so as not to change the usual conditions under which the elderly perform these activities. The ability to climb stairs and wander was evaluated in the presence of the physiotherapist and at places that the elderly residents usually walked.

Based on the information from medical records, we analyzed the following factors: (1) age, (2) sex, (3) height, (4) weight, and (5) body mass index (BMI).

All residents belonged to two nursing homes in the province of León, Castilla y León, Spain. The study population was Caucasian and Spanish-speaking, with a medium–low sociocultural level and a medium economic level. Regarding the religion of the elderly, they were Catholic and the level of education in the majority was basic.

This index has been described by many authors as the most widely used index for evaluating ADL in chronically ill patients and periodically evaluating their progression [27,28,29]. The reliability of the test according to Cronbach’s alpha is 0.86–0.92 for the original version and 0.90–0.92 for the version proposed by Shah et al. [28].

A sample of 68 residents was taken. These people were divided into two groups according to gender (men and women). The inclusion criteria dictated that the patients were older than 65 years [30], that they lived together in the nursing home, and that they had a clinical diagnosis of COVID-19 infection.

The elderly who had contracted and recovered from SARS-CoV-2 were recruited into the study.

The follow-up period was defined as the time that had elapsed from prior to the SARS-CoV-2 infection until recovery. The Barthel index was evaluated over a maximum time frame of six months, a maximum of three months before contracting COVID-19 and a maximum of three months post-infection using the score obtained before infection and after recovery in a certain period of time to evaluate only the impact on the ADL caused by the COVID-19 infection and not the possible events that could happen a posteriori or the improvements derived from rehabilitation.

The protocols used for assessing the elderly were followed at all times according to the guidelines established by the health authorities. Controls were evaluated by polymerase chain reaction (PCR) and antigen tests administered to detect when the virus went into remission and when the patients were transferred to what was called the “clean zone” (the residences had to carry out isolation protocols and divide the buildings into clean and dirty zones, depending on whether the resident had an active infection). Therefore, once the residents obtained a negative result on the tests, they were transferred to the clean zone, free of SARS-CoV-2, and the Barthel assessment was performed. It should be noted that the elderly who went to the COVID-19-free zone had substantially improved their situation and were well enough to be able to resume their pre-infection life, although the majority had limitations.

Indeed, most of the elderly in this study were polymedicated and presented a variety of pathologies. It is true that comorbidities could affect ADL in a manner similar to lung infections suffered in winter (flu, pneumonia, catarrhal processes), which could cause an elderly person to become bed-ridden for days or even weeks. However, the virulence of this infection is devastating, not only for the lives it has claimed, but for the substantial loss of independence for an elderly patient.

Of course, during the days of convalescence, the elderly stopped their physical activities, as happens with other seasonal or bacterial infections, but no infection had caused a loss of muscle function or energy in patients in such a short time as did the COVID-19-induced virus.

The residence was divided into zones, which were delineated by floors, so that residents could move around on the floor on which they were located. Dining and living rooms were doubled so that the elderly could maintain their normal activities as much as possible, but it is true that during the most acute days, as in other infections, the elderly patients remained bedridden.

The habits of the elderly were effectively suspended since the entire operation of the facilities was forced to switch to contingency plans and most types of activities were suspended, which is one reason that the ADLs of the elderly were not the same, but this would not explain such a marked loss in such a short period of time in the ADLs of these patients.

Regarding the issue of comorbidities, we could have conducted a study on whether the comorbidities of the patients studied were a decisive factor in causing the loss of independence when performing ADLs, whether any of the administered medications caused confusion in these patients, or how much the elderly regained their independence in their ADLs once rehabilitation was initiated (it must be taken into account that the elderly began rehabilitation with physiotherapy and occupational therapy after the convalescent period and once discharged with negative PCR results) but the Barthel assessment was administered before rehabilitation to accurately determine the impact of the virus on our participants.

### Statistical Analysis

A descriptive analysis of the characteristics of the participants from both groups was performed. Continuous variables were reported using the mean and standard deviation (SD) and confidence interval 95% (IC95%). The normality of the data was tested using the Shapiro–Wilk test.

For parametric data, paired T-tests were used to determine differences within the same group, and an independent T-test was used between groups.

The differences between before and after COVID-19 were analyzed using one-way repeated-measures analysis of variance (ANOVA). The age, weight, height, and BMI were analyzed as quantitative covariates to test within-subjects effects, and sex was analyzed as a categorical variable to test between-subjects factors, followed by pairwise comparisons using the Bonferroni correction.

For demonstrating the effect size of the comparisons, the Cohen’s d coefficient was calculated. Cohen’s d effect size can be interpreted as described previously: (1) values ≤0.20 indicate slight effects, (2) values between 0.20 and 0.49 indicate fair effects, (3) values between 0.50 and 0.79 indicate moderate effects, and (4) values >0.79 indicate large effects [31].

For all analyses, a value of *p* < 0.05 was considered statistically significant. The data were analyzed using SPSS software for Mac (Version 22; IBM Corp, Armonk, NY, USA).

## 3. Results

All of the variables showed a normal distribution (*p* > 0.05). A significant difference between the ages, heights, and weights of the group of men with respect to the group of women was found; however, for BMI, no significant difference was noted. All data are shown in Table 1.

As can be seen in Table 2, the results of the Barthel index, pre- and post-COVID-19 infection, present significant differences for all evaluated and for the total score.

As can be seen in Table 3, the pre-COVID-19 results of the Barthel index based on gender were compared. Significant results were obtained for both the transfers and ambulation of women compared to men with women, who obtained lower scores for both items; however, after recovering from the COVID-19 infection, the difference in ambulation was still significant between genders. In this case, men obtained worse scores than women, while in transfers, a significant difference did not exist. Urination appeared to be significantly different between men and women, with the latter obtaining the worst scoring, whereas before contracting the infection, differences were insignificant.

## 4. Discussion

In this study, using the Barthel index, the ADLs of patients who had contracted the SARS-CoV-2 infection were evaluated. The Barthel index was applied to evaluate 10 items of ADL in two to four stages; its efficacy is widely accepted for this type of assessment [29,32,33]. The Barthel index has been used to assess functional impairment resulting from multiple sclerosis, cerebrovascular accidents, physical disabilities in the elderly, and many other neurological diseases [29,34,35].

At the end of 2019, a new coronavirus, SARS-CoV-2, began to spread rapidly throughout the world, endangering the health of people around the planet [36]. This new disease causes serious sequelae in 20% of affected patients, and admission to the intensive care unit (ICU) is often necessary due to the respiratory problems; it can even cause death [16].

Muscle weakness is one of the most frequent problems in patients with long bedtime periods and in patients seen in ICUs [37,38]. Critical illness survivors experience marked disability and deficits in physical and cognitive function that can even persist for years after their initial ICU stay [39].

Disability acquired after ICU is associated with reduced health-related quality of life and worse ADL [40].

Our study suggests that ADLs could be reduced after contracting COVID-19. In a study by Iwashyna et al., it was concluded that the elderly, after suffering with severe septicemia, present cognitive impairment and substantial disability that worsens their ability to perform ADLs [41]. Our study shows that COVID-19 infection causes a significant deterioration in all basic ADLs, including eating, washing, dress, getting ready, defecation, urination, using the toilet, transfers, ambulation, and steps, if the results of institutionalized elderly patients with respect to the results of ADL pre- and post-COVID-19 are compared.

Gender and age are the main risk factors for contracting COVID-19 disease [42]. A study found that in similar age groups, the infection was more serious for men than for women [43], and it was men who had the highest mortality rates [44]. These data could explain why, after contracting COVID-19, men obtained worse ADL scores than women despite having a lower average age.

However, the results before contracting the infection were worse for women, a finding that could be explained because aging is the main cause of deterioration [45]. The women in this study had an average age of 87.72 years, which was greater than that of the men; it should be noted that studies conclude that it is from the age of 80 and upward when the death rate of >95% is more significant [46].

The state of emergency resulting from the COVID-19 pandemic, which has had a greater impact on older people, especially with respect to those in institutions, has been minimally studied in terms of the quality of post COVID-19 ADL. Few reports evaluate the relationship between the Barthel index and COVID-19-derived sequelae. We found that the Barthel index, which is a simple and widely used method for assessing ADL, showed a significant correlation with the sequelae suffered by institutionalized patients who had contracted COVID-19, and the total results of the Barthel index could potentially be used to predict the related quality of life after recovering from COVID-19. We believe that the Barthel index is a useful tool for classifying and quantifying impairment in ADLs.

This study has a limitation with respect to the number of participants, but due to the unpredictability of the pandemic, not many elderly people who had undergone a Barthel assessment three months before contracting the COVID-19 infection could be found, regardless of the dramatic mortality caused by the infection (>94%). This situation made it even more difficult to increase the study sample number since it was important to perform the Barthel assessment in a short time frame and ensure that other aspects, such as emerging diseases or improvements due to the rehabilitation of these subjects, did not influence our results.

## 5. Conclusions

Of course, the pandemic has completely taken the worldwide population out of their normal lives, especially the older institutionalized population. It is important to develop maintenance programs for ADLs in situations of this type; perhaps it would be interesting to develop an early rehabilitation plan so that a patient suffering from the infection could continue to undergo rehabilitation despite continuing to test positive on the PCR test in order not to lose ADL capability. This could apply not only to this infection but to all types of illnesses. Promoting the autonomy of patients despite suffering from infection in order to prevent deterioration should be carried out. These pandemic situations must be addressed in order to be prepared and minimize the impact that they may cause on the elderly population.

In summary, this study shows a significant reduction in the quality of ADLs among the elderly institutionalized population in two nursing homes immediately before contracting and after recovering from the COVID-19 infection as measured by the Barthel index.

It appeared that the infection induced by SARS-CoV-2 caused a deterioration in the ADLs more in men than in women and that undoubtedly age was closely related to the loss of the ability to carry out ADLs. Elderly men, especially, saw their abilities diminish more than women, who, although their capacities diminished, did so to a lesser extent.

All of these findings should be taken into account to alleviate the impact that this infection is having not only on the health of our elders and the consequential health expenses that this situation will have but also in terms of the barrier to personal autonomy in the day to day lives of patients. It should be highlighted that the results of this study were not altered by the possible improvement derived from early rehabilitation or the worsening caused by an emerging disease. Data that substantiate our results would be of vital importance for a multidisciplinary team in order to evaluate the deterioration of the ADLs of the surviving elderly people, in order to establish an immediate and personalized rehabilitation plan, not only to preserve their health but also to preserve the quality of life of these people after they recover from this disease.

## Figures and Tables

**Table 1 ijerph-18-07258-t001:** Demographic and descriptive data of the sample population according to male and female groups.

Demographic and Descriptive Data	Total Group N = 68	Male Group N = 34	Female Group N = 34	
	Mean ± SD (IC95%)	Mean ± SD (IC95%)	Mean ± SD (IC95%)	*p*-Value
Age (years)	85.86 ± 6.42	84.00 ± 6.06	87.72 ± 6.34	0.039
	(84.03–87.69)	(81.49–86.50)	(85.09–90.34)	
Weight (Kg)	68.52 ± 14.84	72.76 ± 13.97	64.28 ± 14.74	0.042
	(64.30–72.74)	(66.99–78.52)	(58.19–70.36)	
Height (cm)	168.32 ± 10.85	175.52 ± 9.04	161.12 ± 7.11	0.001
	(165.24–171.40)	(171.78–179.25)	(158.18–164.05)	
BMI (Kg/m^2^)	24.07 ± 4.21	23.54 ± 3.69	24.59 ± 4.68	0.384
	(22.87–25.27)	(22.01–25.07)	(22.65–26.52)	

Abbreviations: BMI, body mass index; SD: standard deviation; IC95%: confidence interval; independent T-tests were used. *p* > 0.05 (with a 95% confidence interval) was considered statistically significant.

**Table 2 ijerph-18-07258-t002:** Pre- and post-Coronavirus 2019 (COVID-19) results based on the Barthel index.

Variables	Before COVID-19 N = 68	After COVID-19 N = 68	*p*-Value	Cohen’s d
	Mean ± DS (IC95%)	Mean ± DS (IC95%)		
Eat	10.00 ± 0.00	8.60 ± 2.48		
	(10.00–10.00)	(7.89–9.31)	<0.001 *	0.37
Wash up	3.00 ± 2.67	1.00 ± 2.02		
	(2.24–3.76)	(0.43–1.57)	<0.001 *	0.38
Dress	8.10 ± 3.18	4.48 ± 4.04		
	(7.20–9.00)	(3.65–5.95)	0.407 *	0.44
Get ready	4.10 ± 1.94	3.10 ± 2.45		
	(3.55–4.65)	(2.40–3.80)	0.490 *	0.22
Deposition	9.60 ± 1.37	7.90 ± 3.51		
	(9.21–9.99)	(6.90–8.90)	<0.001 *	0.30
Urination	8.30 ± 2.60	5.70 ± 3.91		
	(7.56–9.04)	(4.59–6.81)	0.552 *	0.36
Toilet	7.70 ± 4.07	4.30 ± 4.17		
	(6.54–8.86)	(3.12–5.48)	0.600 *	0.38
Transfers	14.50 ± 1.82	9.00 ± 6.14		
	(13.98–15.02)	(7.25–10.75)	<0.001 *	0.51
Ambulation	14.50 ± 1.52	6.60 ± 4.99		
	(14.07–14.93)	(5.18–8.02)	<0.001 *	0.73
Steps	3.30 ± 4.36	0.40 ± 1.98		
	(2.06–4.54)	(0.16–0.96)	<0.001 *	0.39
Total score	83.20 ± 15.20	52.30 ± 27.22		
	(78.7–87.67)	(44.56–60.04)	<0.001 *	0.57

Abbreviations: DS: standard deviation; IC95%: confidence interval; * one-way repeated-measures analysis of variance (ANOVA) was used. *p* < 0.05 (with a 95% confidence interval) was considered statistically significant.

**Table 3 ijerph-18-07258-t003:** Pre- and post-COVID-19 results based on the Barthel index by gender.

	Before COVID-19 N = 68				After COVID-19 N = 68			
	Mean ± DS (IC95%)				Mean ± DS (IC95%)			
Variables	Female	Male	*p*-Value	Cohen’s d	Female	Male	*p*-Value	Cohen’s d
Eat	10.00 ± 0.00	10.00 ± 0.00			8.60 ± 2.29	8.60 ± 2.70		
	(10.00–10.00)	(10.00–10.00)	1.00	NA	(7.65–9.54)	(7.48–9.71)	0.500	0.00
Wash up	2.60 ± 2.54	3.4 ± 2.78			1.20 ± 2.17	0.80 ± 1.87		
	(1.54–3.65)	(2.25–4.54)	0.147	0.13	(7.65–9.54)	(0.02–1.57)	0.244	0.09
Dress	8.00 ± 2.22	8.20 ± 3.18			5.00 ± 4.08	4.60 ± 4.06		
	(6.66–9.33)	(6.88–9.51)	0.413	0.03	(3.31–6.68)	(2.92–6.27)	0.364	0.04
Get ready	4.20 ± 1.87	4.00 ± 2.04			3.00 ± 2.50	3.20 ± 2.44		
	(3.42–4.97)	(3.15–4.84)	0.359	0.05	(1.96–4.03)	(2.18–4.21)	0.388	0.04
Defecation	9.40 ± 1.65	9.80 ± 1.00			7.60 ± 3.85	8.20 ± 3.18		
	(8.71–10.08)	(9.38–10.21)	0.153	0.14	(6.01–9.18)	(6.88–9.51)	0.275	0.08
Urination	8.00 ± 2.50	8.60 ± 2.70			4.6 ± 3.79	6.80 ± 3.78		
	(6.96–9.03)	(7.48–9.71)	0.209	0.11	(3.03–6.16)	(5.23–8.36)	0.022 *	0.27
Toilet	7.40 ± 4.11	8.00 ± 4.08			4.00 ± 4.33	4.60 ± 4.06		
	(5.70–9.09)	(6.31–9.68)	0.303	0.07	(2.21–5.78)	(2.92–6.27)	0.307	0.07
Transfers	14.00 ± 2.50	14.96 ± 0.20			9.20 ± 5.71	8.80 ± 6.65		
	(12.96–15.03)	(14.87–15.04)	0.031 *	0.26	(6.84–11.55)	(6.06–11.54)	0.410	0.03
Ambulation	14.00 ± 2.04	14.96 ± 0.20			8.40 ± 4.72	4.80 ± 4.67		
	(13.15–14.84)	(14.87–15.04)	0.012 *	0.31	(4.44–10.35)	(2.87–6.72)	0.004 *	0.35
Steps	3.00 ± 4.08	3.60 ± 4.68			0.80 ± 2.76	0.00 ± 0.00		
	(1.31–4.68)	(1.66–5.53)	0.315	0.07	(−0.3–1.94)	(0.00–0.00)	0.077	0.20
Total score	80.80 ± 16.18	85.60 ± 15.22			53.00 ± 28.43	51.60 ± 26.52		
	(74.12–87.47)	(79.31–91.88)	0.142	0.15	(41.26–64.73)	(40.65–62.54)	0.428	0.02

Abbreviations: DS: standard deviation; IC95%: confidence interval; NA: not applicable; * independent T-tests were used. *p* < 0.05 (with a 95% confidence interval) was considered statistically significant.

## Data Availability

The dataset supporting the conclusions of this article is available in the marta.losa@urjc.es in the Faculty of Health Sciences, Universidad Rey Juan Carlos, 28933 Madrid, Spain.

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
