# Peer review of "Use of the Barthel Index to Assess Activities of Daily Living before and after SARS-COVID 19 Infection of Institutionalized Nursing Home Patients"

_ijerph, 2021, doi:10.3390/ijerph18147258_

Round 1

Reviewer 1 Report

Thank you for the opportunity to read your manuscript entitled "Use of the Barthel index to assess activities of daily living before and after SARS-COVID19 infection of institutionalized nursing home patients". I have some comments that would improve the quality of the manuscript below:

Major revisions:

  1. Please include a section in the methods of: 1) why you selected the Barthel index and 2) the validity and reliability of this index in individuals that are living in an institutionalized nursing home".
  2. It's my understanding that you measured ADLs in residents prior to COVID? If this is true, could you please clarify this in the methods with a timeline.
  3. Why did the authors perform a two-tailed independent t-test? I believe a one-tailed test would be more accurate?
  4. Cochrane suggests we collect demographic data beyond just age, sex, weight and height (PROGRESS-PLUS) - Can the other's include more information about the residents using PROGRESS-PLUS (https://methods.cochrane.org/equity/projects/evidence-equity/progress-plus)
  5. It's unclear to me when the index was used "after COVID infection". Did the authors wait till the residents were feeling better? And if so, how long after and was this the same for all residents or did some residents have more time to recover than others? How did the authors account for long-haulers. The timeline could use clarification.
  6. Is it possible to include medical history and medications to determine if co-morbidity also affected ability to perform ADLs. Were the residents previously exercising and then stopped due to COVID-19 which could affect muscle strength. I don't think the authors took into account confounding factors, such has possibly new co-morbidities or medications that may have been present/not present between the two measures. Another factor could be a result of longer bedtime if residences could not leave their rooms because of COVID-19. If the authors could provide a descriptions of the institution, how outbreaks were handled, how outbreaks may have affected mobility of participants, which could have affected ADLs because of muscle loss due to prolonged inactivity. Did residents' habits change and could that have affected ADLs.

Minor revisions

  1. I suggest using period instead of commas to indicate decimal positions
  2. I would avoid making direct conclusions. For example in the discussion the authors state " Our study reveals that activities of daily living are reduced after suffering COVID-19" - however, I do not think this statement is valid without taking into account other factors such as prolonged bedrest/immobility (i.e., were residents asked to stay in their rooms for longer than usual because of COVID-19), medical history, etc.

Author Response

Comments and Suggestions for Authors

1.Please include a section in the methods of: 1) why you selected the Barthel index and 2) the validity and reliability of this index in individuals that are living in an institutionalized nursing home".

In the homes for the elderly in Spain, more specifically in those of Castilla y León, the community in which our study was carried out, the activities of daily life of the residents to establish the degree of dependency, is established by ROYAL DECREE 504 / 2007, of April 20, by the that the assessment scale of the dependency situation established by the Law 39/2006, of December 14, on promotion of personal autonomy and care for people in a situation of dependency. Based on the decree, it is mandatory for nursing homes to carry out at least 2 determinations of the Barthel index in residences annually, therefore based on this regional law and that the Barthel Index has a track record of being used in numerous studies and is still widely used today as a simple method to assess ADL for various diseases (Mahoney FI), we decided that it could be a good indicator to see the impact that the pandemic had had on our institutionalized elders.

2.It's my understanding that you measured ADLs in residents prior to COVID? If this is true, could you please clarify this in the methods with a timeline.

The residences that were taken as a sample are among those with the greatest number of elderly people, and they are the residences that carry out more control over the activities of daily living because they have among their places some of them arranged (this means that part of the cats derived from living in the residence are publicly paid for) the administration does a very exhaustive monitoring of the places it has arranged, and this residence performs Barthel indexes every 3 months, that is why, when the pandemic hit the residence, all the elderly had been tested with the index at most 3 months before, and taking into account that Sars-Cov-2 entered the residence, affecting almost all the residents in a short period of time, we were able to perform the Barthel indexes in the survivors just after suffering the virus, with a maximum post-exposure time of 3 months, thus assessing the pure effects of the virus without the possibility of improvement after rehabillitation, derived diseases, complications etc

3.Why did the authors perform a two-tailed independent t-test? I believe a one-tailed test would be more accurate?

We use 2 tailed test for simple size calculation.

  1. Cochrane suggests we collect demographic data beyond just age, sex, weight and height (PROGRESS-PLUS) - Can the other's include more information about the residents using PROGRESS-PLUS (https://methods.cochrane.org/equity/projects/evidence-equity/progress-plus)

All residents belong to 2 nursing homes in the province of León, Castilla y León, Spain. The elderly studied are Caucasian, all Spanish-speaking, with a medium-low socio-cultural level and a medium economic level. Regarding the religion of the elderly they are Catholic and the level of studies in the majority is basic.

  1. It's unclear to me when the index was used "after COVID infection". Did the authors wait till the residents were feeling better? And if so, how long after and was this the same for all residents or did some residents have more time to recover than others? How did the authors account for long-haulers. The timeline could use clarification.

The protocols that were followed with the elderly were determined at all times by what was established by the health authorities, controls were carried out by PCR and antigen tests to the elderly to detect when the virus had remitted and it was when they were transferred to what was called “clean zone” (the residences had to carry out isolation protocols and divide the residences into clean and dirty zones, depending on whether the infection was found in the residents). Therefore, once the residents gave negative in the tests, they were transferred to the clean zone free of Sars-Cov-2 and the Barthel was carried out. It should be noted that the elderly who went to the covid-free zone had substantially improved their situation and were better enough to be able to resume their life before the infection, although the majority had limitations.

  1. Is it possible to include medical history and medications to determine if co-morbidity also affected ability to perform ADLs. Were the residents previously exercising and then stopped due to COVID-19 which could affect muscle strength. I don't think the authors took into account confounding factors, such has possibly new co-morbidities or medications that may have been present/not present between the two measures. Another factor could be a result of longer bedtime if residences could not leave their rooms because of COVID-19. If the authors could provide a descriptions of the institution, how outbreaks were handled, how outbreaks may have affected mobility of participants, which could have affected ADLs because of muscle loss due to prolonged inactivity. Did residents' habits change and could that have affected ADLs.

Indeed, most of the elderly under study are polymedicated and present a variety of pathologies, and it is true that comorbidity could affect ADL, but like lung infections suffered in winter (flu, pneumonia, catarrhal processes, etc.) they prostrate to the elderly in bed for days and even weeks, SARS-cov-2 has done it, and what has been observed is that the virulence of this infection has been devastating, not only for the lives it has claimed, but for the substantial loss of independence that the elderly had.

Of course, during the days of convalescence, the elderly stopped their physical activity, as happens with other seasonal or bacterial infections, but no infection had diminished in such a short time and in the same way the muscles and energy of these patients .

The residence was divided into zones, which were delimited by its floors, so that residents could move around the floor they were on, dining rooms and living rooms were doubled, so that the elderly could maintain their activity as much as possible. normal possible, but it is true that during the most acute days, and as in other infections, the elderly remained in bed.

The habits of the elderly effectively changed, since the entire operation of the residences, carrying out contingency plans at a forced march and suspending activities that’s why inevitably the activities of daily life of the elderly were not the same, but this would not explain the loss so marked and in such a long way. little time of the AVD.

What you ask us about the issue of comorbidity is very interesting, we could carry out a study on whether the comorbidity of the patients studied is decisive in the loss of independence when performing ADL, if any of the administered medications affects confusion of the elderly or on how and how much the elderly regain independence in their ADL once the rehabilitation began (it must be taken into account that the elderly after the convalescent period and once discharged with negative CRP, began rehabilitation with the service physiotherapy and occupational therapy) but the Barthel did them before to determine as pure as possible the impact of the virus on our elders.

Minor revisions

  1. I suggest using period instead of commas to indicate decimal positions

We remove commas

  1. I would avoid making direct conclusions. For example in the discussion the authors state " Our study reveals that activities of daily living are reduced after suffering COVID-19" - however, I do not think this statement is valid without taking into account other factors such as prolonged bedrest/immobility (i.e., were residents asked to stay in their rooms for longer than usual because of COVID-19), medical history, etc.

We changed to:

Our study suggests that activities of daily living could be reduced after suffering COVID-19

Comments and Suggestions for Authors

1.Please include a section in the methods of: 1) why you selected the Barthel index and 2) the validity and reliability of this index in individuals that are living in an institutionalized nursing home".

In the homes for the elderly in Spain, more specifically in those of Castilla y León, the community in which our study was carried out, the activities of daily life of the residents to establish the degree of dependency, is established by ROYAL DECREE 504 / 2007, of April 20, by the that the assessment scale of the dependency situation established by the Law 39/2006, of December 14, on promotion of personal autonomy and care for people in a situation of dependency. Based on the decree, it is mandatory for nursing homes to carry out at least 2 determinations of the Barthel index in residences annually, therefore based on this regional law and that the Barthel Index has a track record of being used in numerous studies and is still widely used today as a simple method to assess ADL for various diseases (Mahoney FI), we decided that it could be a good indicator to see the impact that the pandemic had had on our institutionalized elders.

2.It's my understanding that you measured ADLs in residents prior to COVID? If this is true, could you please clarify this in the methods with a timeline.

The residences that were taken as a sample are among those with the greatest number of elderly people, and they are the residences that carry out more control over the activities of daily living because they have among their places some of them arranged (this means that part of the cats derived from living in the residence are publicly paid for) the administration does a very exhaustive monitoring of the places it has arranged, and this residence performs Barthel indexes every 3 months, that is why, when the pandemic hit the residence, all the elderly had been tested with the index at most 3 months before, and taking into account that Sars-Cov-2 entered the residence, affecting almost all the residents in a short period of time, we were able to perform the Barthel indexes in the survivors just after suffering the virus, with a maximum post-exposure time of 3 months, thus assessing the pure effects of the virus without the possibility of improvement after rehabillitation, derived diseases, complications etc

3.Why did the authors perform a two-tailed independent t-test? I believe a one-tailed test would be more accurate?

We use 2 tailed test for simple size calculation.

  1. Cochrane suggests we collect demographic data beyond just age, sex, weight and height (PROGRESS-PLUS) - Can the other's include more information about the residents using PROGRESS-PLUS (https://methods.cochrane.org/equity/projects/evidence-equity/progress-plus)

All residents belong to 2 nursing homes in the province of León, Castilla y León, Spain. The elderly studied are Caucasian, all Spanish-speaking, with a medium-low socio-cultural level and a medium economic level. Regarding the religion of the elderly they are Catholic and the level of studies in the majority is basic.

  1. It's unclear to me when the index was used "after COVID infection". Did the authors wait till the residents were feeling better? And if so, how long after and was this the same for all residents or did some residents have more time to recover than others? How did the authors account for long-haulers. The timeline could use clarification.

The protocols that were followed with the elderly were determined at all times by what was established by the health authorities, controls were carried out by PCR and antigen tests to the elderly to detect when the virus had remitted and it was when they were transferred to what was called “clean zone” (the residences had to carry out isolation protocols and divide the residences into clean and dirty zones, depending on whether the infection was found in the residents). Therefore, once the residents gave negative in the tests, they were transferred to the clean zone free of Sars-Cov-2 and the Barthel was carried out. It should be noted that the elderly who went to the covid-free zone had substantially improved their situation and were better enough to be able to resume their life before the infection, although the majority had limitations.

  1. Is it possible to include medical history and medications to determine if co-morbidity also affected ability to perform ADLs. Were the residents previously exercising and then stopped due to COVID-19 which could affect muscle strength. I don't think the authors took into account confounding factors, such has possibly new co-morbidities or medications that may have been present/not present between the two measures. Another factor could be a result of longer bedtime if residences could not leave their rooms because of COVID-19. If the authors could provide a descriptions of the institution, how outbreaks were handled, how outbreaks may have affected mobility of participants, which could have affected ADLs because of muscle loss due to prolonged inactivity. Did residents' habits change and could that have affected ADLs.

Indeed, most of the elderly under study are polymedicated and present a variety of pathologies, and it is true that comorbidity could affect ADL, but like lung infections suffered in winter (flu, pneumonia, catarrhal processes, etc.) they prostrate to the elderly in bed for days and even weeks, SARS-cov-2 has done it, and what has been observed is that the virulence of this infection has been devastating, not only for the lives it has claimed, but for the substantial loss of independence that the elderly had.

Of course, during the days of convalescence, the elderly stopped their physical activity, as happens with other seasonal or bacterial infections, but no infection had diminished in such a short time and in the same way the muscles and energy of these patients .

The residence was divided into zones, which were delimited by its floors, so that residents could move around the floor they were on, dining rooms and living rooms were doubled, so that the elderly could maintain their activity as much as possible. normal possible, but it is true that during the most acute days, and as in other infections, the elderly remained in bed.

The habits of the elderly effectively changed, since the entire operation of the residences, carrying out contingency plans at a forced march and suspending activities that’s why inevitably the activities of daily life of the elderly were not the same, but this would not explain the loss so marked and in such a long way. little time of the AVD.

What you ask us about the issue of comorbidity is very interesting, we could carry out a study on whether the comorbidity of the patients studied is decisive in the loss of independence when performing ADL, if any of the administered medications affects confusion of the elderly or on how and how much the elderly regain independence in their ADL once the rehabilitation began (it must be taken into account that the elderly after the convalescent period and once discharged with negative CRP, began rehabilitation with the service physiotherapy and occupational therapy) but the Barthel did them before to determine as pure as possible the impact of the virus on our elders.

Minor revisions

  1. I suggest using period instead of commas to indicate decimal positions

We remove commas

  1. I would avoid making direct conclusions. For example in the discussion the authors state " Our study reveals that activities of daily living are reduced after suffering COVID-19" - however, I do not think this statement is valid without taking into account other factors such as prolonged bedrest/immobility (i.e., were residents asked to stay in their rooms for longer than usual because of COVID-19), medical history, etc.

We changed to:

Our study suggests that activities of daily living could be reduced after suffering COVID-19

Comments and Suggestions for Authors

1.Please include a section in the methods of: 1) why you selected the Barthel index and 2) the validity and reliability of this index in individuals that are living in an institutionalized nursing home".

In the homes for the elderly in Spain, more specifically in those of Castilla y León, the community in which our study was carried out, the activities of daily life of the residents to establish the degree of dependency, is established by ROYAL DECREE 504 / 2007, of April 20, by the that the assessment scale of the dependency situation established by the Law 39/2006, of December 14, on promotion of personal autonomy and care for people in a situation of dependency. Based on the decree, it is mandatory for nursing homes to carry out at least 2 determinations of the Barthel index in residences annually, therefore based on this regional law and that the Barthel Index has a track record of being used in numerous studies and is still widely used today as a simple method to assess ADL for various diseases (Mahoney FI), we decided that it could be a good indicator to see the impact that the pandemic had had on our institutionalized elders.

2.It's my understanding that you measured ADLs in residents prior to COVID? If this is true, could you please clarify this in the methods with a timeline.

The residences that were taken as a sample are among those with the greatest number of elderly people, and they are the residences that carry out more control over the activities of daily living because they have among their places some of them arranged (this means that part of the cats derived from living in the residence are publicly paid for) the administration does a very exhaustive monitoring of the places it has arranged, and this residence performs Barthel indexes every 3 months, that is why, when the pandemic hit the residence, all the elderly had been tested with the index at most 3 months before, and taking into account that Sars-Cov-2 entered the residence, affecting almost all the residents in a short period of time, we were able to perform the Barthel indexes in the survivors just after suffering the virus, with a maximum post-exposure time of 3 months, thus assessing the pure effects of the virus without the possibility of improvement after rehabillitation, derived diseases, complications etc

3.Why did the authors perform a two-tailed independent t-test? I believe a one-tailed test would be more accurate?

We use 2 tailed test for simple size calculation.

  1. Cochrane suggests we collect demographic data beyond just age, sex, weight and height (PROGRESS-PLUS) - Can the other's include more information about the residents using PROGRESS-PLUS (https://methods.cochrane.org/equity/projects/evidence-equity/progress-plus)

All residents belong to 2 nursing homes in the province of León, Castilla y León, Spain. The elderly studied are Caucasian, all Spanish-speaking, with a medium-low socio-cultural level and a medium economic level. Regarding the religion of the elderly they are Catholic and the level of studies in the majority is basic.

  1. It's unclear to me when the index was used "after COVID infection". Did the authors wait till the residents were feeling better? And if so, how long after and was this the same for all residents or did some residents have more time to recover than others? How did the authors account for long-haulers. The timeline could use clarification.

The protocols that were followed with the elderly were determined at all times by what was established by the health authorities, controls were carried out by PCR and antigen tests to the elderly to detect when the virus had remitted and it was when they were transferred to what was called “clean zone” (the residences had to carry out isolation protocols and divide the residences into clean and dirty zones, depending on whether the infection was found in the residents). Therefore, once the residents gave negative in the tests, they were transferred to the clean zone free of Sars-Cov-2 and the Barthel was carried out. It should be noted that the elderly who went to the covid-free zone had substantially improved their situation and were better enough to be able to resume their life before the infection, although the majority had limitations.

  1. Is it possible to include medical history and medications to determine if co-morbidity also affected ability to perform ADLs. Were the residents previously exercising and then stopped due to COVID-19 which could affect muscle strength. I don't think the authors took into account confounding factors, such has possibly new co-morbidities or medications that may have been present/not present between the two measures. Another factor could be a result of longer bedtime if residences could not leave their rooms because of COVID-19. If the authors could provide a descriptions of the institution, how outbreaks were handled, how outbreaks may have affected mobility of participants, which could have affected ADLs because of muscle loss due to prolonged inactivity. Did residents' habits change and could that have affected ADLs.

Indeed, most of the elderly under study are polymedicated and present a variety of pathologies, and it is true that comorbidity could affect ADL, but like lung infections suffered in winter (flu, pneumonia, catarrhal processes, etc.) they prostrate to the elderly in bed for days and even weeks, SARS-cov-2 has done it, and what has been observed is that the virulence of this infection has been devastating, not only for the lives it has claimed, but for the substantial loss of independence that the elderly had.

Of course, during the days of convalescence, the elderly stopped their physical activity, as happens with other seasonal or bacterial infections, but no infection had diminished in such a short time and in the same way the muscles and energy of these patients .

The residence was divided into zones, which were delimited by its floors, so that residents could move around the floor they were on, dining rooms and living rooms were doubled, so that the elderly could maintain their activity as much as possible. normal possible, but it is true that during the most acute days, and as in other infections, the elderly remained in bed.

The habits of the elderly effectively changed, since the entire operation of the residences, carrying out contingency plans at a forced march and suspending activities that’s why inevitably the activities of daily life of the elderly were not the same, but this would not explain the loss so marked and in such a long way. little time of the AVD.

What you ask us about the issue of comorbidity is very interesting, we could carry out a study on whether the comorbidity of the patients studied is decisive in the loss of independence when performing ADL, if any of the administered medications affects confusion of the elderly or on how and how much the elderly regain independence in their ADL once the rehabilitation began (it must be taken into account that the elderly after the convalescent period and once discharged with negative CRP, began rehabilitation with the service physiotherapy and occupational therapy) but the Barthel did them before to determine as pure as possible the impact of the virus on our elders.

Minor revisions

  1. I suggest using period instead of commas to indicate decimal positions

We remove commas

  1. I would avoid making direct conclusions. For example in the discussion the authors state " Our study reveals that activities of daily living are reduced after suffering COVID-19" - however, I do not think this statement is valid without taking into account other factors such as prolonged bedrest/immobility (i.e., were residents asked to stay in their rooms for longer than usual because of COVID-19), medical history, etc.

We changed to:

Our study suggests that activities of daily living could be reduced after suffering COVID-19

Reviewer 2 Report

Introduction

The goal of the study needs to be properly highlighted and justified. Instead of setting their aim in the frame of a simple question, I would recommend that the authors attempt to present the key objectives of their study with regards to what is presently known (i.e. literature), thus highlighting the added value of the article.

Methods
Could the authors please add information on how the participants were recruited?

How normality was tested?

Please report cohen's d data

Conclusion
It would be appreciated if the authors could give more details about implications for prevention and intervention.

I recommend the author review the English of this manuscript and reformulate tables and figures to be more clear to the audience

Author Response

Introduction

The goal of the study needs to be properly highlighted and justified. Instead of setting their aim in the frame of a simple question, I would recommend that the authors attempt to present the key objectives of their study with regards to what is presently known (i.e. literature), thus highlighting the added value of the article.

We wrote:

Therefore, the goal of this study was to evaluate the ADL in residents before and after suffering COVID-19

Methods
Could the authors please add information on how the participants were recruited?

The elderly who suffered and exceeded Sars-cov-2 were recruited.

How normality was tested?

he normality of the data were tested using Shapiro-Wilk test.

Please report cohen's d data

For demonstrating the effect size of the comparisons, the Cohen d coefficient was calculated.Cohen´s d effect size can be interpreted as follows: values ≤0.20 indicate slight effects, values between 0.20 and 0.49 indicate fair effects, values between 0.50 and 0.79 indicate moderate effects, and values larger than 0.79 indicate large effects [ Cohen J. A power primer. Psychol Bull 1992;112:155–9].

We insert the cohen´s data in table 2 and table 3

Conclusion

It would be appreciated if the authors could give more details about implications for prevention and intervention.

Of course, the pandemic has completely taken the world's population out of the game, and especially the older population that lives institutionalized. It would be important to develop maintenance programs for activities of daily living in situations of this type, perhaps it would be interesting to develop an early rehabilitation plan so that the patient suffering from the infection could continue to rehabilitate despite continuing to test positive in the PCR with the In order not to lose capabilities, but not only with this infection but with all of them in all areas.

Promote the autonomy of patients despite suffering from the infection in order to stop the deterioration. These pandemic situations must be addressed in order to be prepared and minimize the impact they may cause on the elderly.

I recommend the author review the English of this manuscript and reformulate tables and figures to be more clear to the audience

We sent the manuscript to AME editors

Reviewer 3 Report

In the introduction section, it would be convenient to provide epidemiological data on covid infection in nursing homes for the elderly, at least in Spain.

It is necessary for you to draw conclusions from what is found in relation to sex. Investigating if there are differences by sex is part of the objective.

What average time passed since Barthel's pre-covid and post-covid assessment? You must reflect it in the methodology section

Author Response

In the introduction section, it would be convenient to provide epidemiological data on covid infection in nursing homes for the elderly, at least in Spain.

In Spain there are 5,417 nursing homes, 690 in Castilla y León, of which 71% are private (García et al.)

According to the recent monographic report on Spain from the ltccovid.org portal, which belongs to the International Long Term Care Policy Network, a network managed by the London School of Economics (LSE), data updated on May 28, indicate 237,906 infected by COVID- 19 in Spain and 27,119 died from this disease. Deaths in nursing homes would rise to 19,194, 70% of the total number of deaths, 2,449 in Castilla y León (Zalakaín et al.)

It is necessary for you to draw conclusions from what is found in relation to sex. Investigating if there are differences by sex is part of the objective.

In summary, it seems that the infection by Sars-Cov-2 deteriorates the activities of daily life more in men than in women and that undoubtedly age is closely related to the loss of the ability to carry out activities of daily living, being to the elderly, older men, those who would see their abilities diminished more against women, who, although they see their capacities diminished, do so to a lesser extent.

What average time passed since Barthel's pre-covid and post-covid assessment? You must reflect it in the methodology section

The barthel indices were performed in a maximum time frame of 6 months, a maximum of 3 months before suffering covid, and a maximum of 3 months after suffering covid.

Round 2

Reviewer 1 Report

This type of design should use a one-tailed test because of the direction of the Barthel index. In addition the authors should have selected a number of variables a priori to adjust for in the calculations. The authors will need to be clear that the selection of adjusted variables such as age, sex, # of co-morbidies and physical activity level were selected post hoc and adjust for these variables in their calculation. Please use 95% Confidence intervals instead of SD since it provides the readers with a lot more information.

Author Response

This type of design should use a one-tailed test because of the direction of the Barthel index. In addition the authors should have selected a number of variables a priori to adjust for in the calculations. The authors will need to be clear that the selection of adjusted variables such as age, sex, # of co-morbidies and physical activity level were selected post hoc and adjust for these variables in their calculation.  Please use 95% Confidence intervals instead of SD since it provides the readers with a lot more information.

Answer:

Thanks a lot to the reviewer. We performed the test you suggested

The differences between before and after COVID-19 were analyzed using one-way repeated-measures analysis of variance (ANOVA). The age, weight, Height and BMI were analyzed as quantitative covariates to test Within-Subjects Effects and sex were analyzed as categorical variable to test Between-Subjects Factors, followed by pairwise comparisons using the Bonferroni correction.

We removed the words “range” due to IC95% were calculated.

We sent the article to AME to copyedit the text.

Reviewer 2 Report

The article is now suitable for publication 

Author Response

Answer:

Thanks a lot to the reviewer.

We sent the article to American Manuscript Editors to copyedit the text.

This manuscript is a resubmission of an earlier submission. The following is a list of the peer review reports and author responses from that submission.